# Close Relationship between Systemic Arterial and Portal Venous Pressure in an Animal Model with Healthy Liver

**DOI:** 10.3390/ijms24129963

**Published:** 2023-06-09

**Authors:** Adhara Lazaro, Patrick Stoll, Dominik von Elverfeldt, Wolfgang Kreisel, Peter Deibert

**Affiliations:** 1Institute of Exercise and Occupational Medicine, Faculty of Medicine, Medical Center, University of Freiburg, 79106 Freiburg, Germany; adhara.lazaro@alumni.uni-heidelberg.de (A.L.); peter.deibert@uniklinik-freiburg.de (P.D.); 2Narkose Suedbaden, 79110 Freiburg, Germany; stoll@narkose-suedbaden.de; 3Department of Diagnostic and Interventional Radiology, Division of Medical Physics, Faculty of Medicine, Medical Center, University of Freiburg, 79106 Freiburg, Germany; dominik.elverfeldt@uniklinik-freiburg.de; 4Department of Medicine II, Gastroenterology, Hepatology, Endocrinology and Infectious Diseases, Faculty of Medicine, Medical Center, University of Freiburg, 79106 Freiburg, Germany

**Keywords:** portal pressure, systemic arterial pressure, portal hypertension, liver cirrhosis, sildenafil, phosphodiesterase, hemodynamics

## Abstract

It is unclear to what extent systemic arterial blood pressure influences portal pressure. This relationship is clinically important as drugs, which are conventionally used for therapy of portal hypertension, may also influence systemic arterial blood pressure. This study investigated the potential correlation between mean arterial (MAP) and portal venous pressure (PVP) in rats with healthy livers. In a rat model with healthy livers, we investigated the effect of manipulation of MAP on PVP. Interventions consisted of 0.9% NaCl (group 1), 0.1 mg/kg body weight (bw) Sildenafil (low dose), an inhibitor of phosphodiesterase-5 (group 2), and 1.0 mg/kg bw Sildenafil (high dose, group 3) in 600 µL saline injected intravenously. Norepinephrine was used to increase MAP in animals with circulatory failure while PVP was monitored. Injection of the fluids induced a transient drop in MAP and PVP, probably due to a reversible cardiac decompensation. The drop in MAP and drop in PVP are significantly correlated. The time lag between change in MAP and change in PVP by 24 s in all groups suggests a cause-and-effect relationship. Ten minutes after the injection of the fluid, cardiac function was normalized. Thereafter, MAP gradually decreased. In the NaCl group, PVP decreases by 0.485% for a 1% drop of MAP, by 0.550% in the low-dose sildenafil group, and by 0.651% in the high-dose sildenafil group (*p* < 0.05 for difference group two vs. group one, group three vs. group one, and group three vs. group two). These data suggest that Sildenafil has an inherent effect on portal pressure that exceeds the effect of MAP. Injection of norepinephrine led to a sudden increase in MAP followed by an increase in PVP after a time lag. These data show a close relationship between portal venous pressure and systemic arterial pressure in this animal model with healthy livers. A change in MAP is consequently followed by a change in PVP after a distinct time lag. This study, furthermore, suggests that Sildenafil influences portal pressure. Further studies should be performed in a model with cirrhotic livers, as these may be important in the evaluation of vasoactive drugs (e.g., PDE-5-inhibitors) for therapy of portal hypertension.

## 1. Introduction

Portal hypertension is defined as portal venous pressure exceeding 5 mm Hg [1,2,3,4]. If portal pressure exceeds 10–12 mm Hg, life-threatening complications such as esophageal variceal bleeding, ascites, hepatic encephalopathy, hepato-renal syndrome, hepato-pulmonary syndrome, or porto-pulmonary hypertension may occur [5,6,7,8].

Depending on the underlying cause, PH can be classified as prehepatic, intrahepatic, and post-hepatic [1,9,10,11]. The most frequent cause of intrahepatic PH is liver cirrhosis. A structural component—the disturbed liver architecture, fibrotic tissue, regenerative nodules, neoangiogenesis, narrowing of sinusoids, and vascular occlusion—causes about 70% of the intrahepatic vascular resistance. In addition, dynamic components related to the interplay of sinusoidal cells, including hepatocytes (HCs), liver sinusoidal endothelial cells (LSECs), and hepatic stellate cells (HSCs) account for the remaining 30%. These dynamics lead to increased sinusoidal tone due to an imbalance of dilating and constricting factors [1,12,13,14,15,16,17,18,19]. Lastly, an increased splanchnic blood flow further contributes to the rise in portal pressure [20,21,22,23,24].

One critical component in the pathogenesis of PH is the altered regulation of nitric oxide (NO) in the vascular system. NO plays a key role in the regulation of vascular tone, with its availability being reduced in the liver, leading to sinusoidal constriction, while outside the liver, NO availability increases, leading to peripheral vessel dilation [16,25,26]. An opposed regulation of cyclic guanosine monophosphate (cGMP) may explain this disturbed regulation, with overexpressed phosphodiesterase-5 (PDE-5) leading to degradation of cGMP in the sinusoids (sinusoidal constriction), while its concentration in the periphery increases due to low PDE-5 expression (peripheral vessel dilation) [27].

Therapeutic strategies for PH focus on lowering portal pressure to prevent the onset of complications [8,28,29,30,31]. Non-selective beta-blockers (NSBB), particularly propranolol, are the most commonly used drugs for treatment of PH. Carvedilol (a NSBB with anti-alpha-1-receptor activity) is now the preferred NSBB [8]. Organic nitrates may be used in combination with NSBB. Other vasoactive drugs have been investigated as potential medical therapies of PH, such as ACE-inhibitors, AT-1-receptor antagonists, and prazosin (an alpha-1-receptor blocker). PDE-5 inhibitors such as sildenafil, which inhibit the conversion of cGMP to inactive 5’-GMP, have emerged as a novel approach to manage PH, addressing the disturbed regulation of the NO–cGMP system within the sinusoids [27,32,33,34,35].

However, the systemic effects of these treatments pose a challenge. Every currently used medical therapy of PH also has some influence on systemic circulation [36,37,38,39,40,41,42]. The degree to which the drop in systemic blood pressure versus the specific effect on portal pressure contributes to the efficacy of PH therapy remains unclear. Clinical studies to monitor portal pressure in relation to changes in systemic blood pressure in humans are ethically challenging. Very few studies investigated the effect of blood loss and volume substitution in animals with liver cirrhosis [43,44]. An animal study in which a modulation of systemic blood pressure and its effect on portal blood pressure were investigated has not been performed yet.

Considering the gap in our understanding of how systemic hemodynamics impact portal pressure, particularly in the context of PH, there is a critical need for comprehensive studies. Investigations that focus on understanding the pathophysiology of PH, particularly the role of the NO–cGMP pathway, and the effects of blood pressure modulation, are essential to improve the therapeutic management of PH. Such studies would not only provide a deeper understanding of the complex dynamics of PH but also pave the way for the development of targeted therapeutics, optimizing the balance between systemic and portal effects.

Our previous research focused on the role of key enzymes in the NO–cGMP pathway in healthy rats and cirrhosis induced by thioacetamide (TAA) [33]. The present paper aims to elaborate on our previous work, further exploring the relationship between systemic and portal hemodynamics in healthy rats. Specifically, we aim to investigate the correlation between MAP and PVP in greater depth. Once we better understand the complex relationship between systemic and portal pressure, we can move on to an in-depth analysis of pathophysiological changes in liver cirrhosis and to a better assessment of the potential of PDE-5 inhibitors in managing PH.

## 2. Results

Complete hemodynamic analysis was performed in 15 rats given 0.9% NaCl (group one), 18 rats given sildenafil 0.1 mg/kg bw in 0.9% NaCl (group two), and 18 rats given sildenafil 1 mg/kg bw in 0.9% NaCl (group three). Baseline characteristics (Table 1) show no significant difference between the groups. In three animals, circulatory failure occurred and administration of norepinephrine was necessary. One rat expired due to perforation of the portal vein and blood loss. These four rats were excluded from further data analyses.

### 2.1. Time-Dependent Course of MAP and PVP in Three Groups of Healthy Rats

Figure 1 shows the course of MAP and PVP in each group over time after injection of 600 µL fluid in the superior vena cava. The simultaneous measurement of the two parameters, MAP and PVP, allowed conclusions to be drawn about their correlation.

After central intravenous application of the study drug or placebo (i.e., sildenafil, NaCl), there is a dip/drop in both parameters, MAP and PVP, due to transient cardiac decompensation secondary to fluid overload. Heart rate shows a similar dip, further suggesting artificial and transient cardiac decompensation during the first minutes. After a few minutes, cardiac function normalizes and the effect of the substance becomes evident.

For further analyses, different time intervals were evaluated. We focused on correlation between MAP and PVP, as well as a possible effect of sildenafil on portal venous pressure.

### 2.2. Correlation between MAP and PVP

To quantify the correlation between the mean of MAP and PVP, random effects regression analyses with logarithmic outcome and explanatory variables (log–log specification) were used. The logarithmic specification allows the analysis of the existence of correlation, as well as quantifying it in meaningful terms (Table 2).

### 2.3. Evaluation of Time Interval 10–30 min (Excluding the “Dip”)

The dip reflects the deterioration of cardiac function after the administration of 600 µL fluid. This might have confounded the correlation of the two parameters. Ten minutes after injection, the cardiac function returns to normal. Therefore, we analyzed the course of MAP and PVP over a period of 10 to 30 min after the intervention using regression analysis in order to find a possible correlation of the parameters. The coefficients (change in PVP in relation to change in MAP) were calculated (Table 2). In group one (NaCl), PVP decreases/increases by 0.485% for every 1% decrease/increase in MAP. In group two (sildenafil 0.1 mg/kg), PVP decreases/increases by 0.550% for every 1% decrease/increase in MAP. Moreover, in the third group (sildenafil 1 mg/kg), PVP decreases/increases by 0.651% for every 1% decrease/increase in MAP. These data suggest a correlation between MAP and PVP during the time interval 10–30 min. The three groups are significantly different from each other (*p* < 0.001) with respect to the relationship between ln MAP and ln PVP, as is also confirmed using *t*-test.

In addition, injection of sildenafil has a dose-dependent effect on the decrease in portal pressure in comparison to saline.

### 2.4. Correlation between the Maximum Dip in MAP and the Maximum Dip in PVP during the First 10 min

If there is an interdependent relationship between MAP and PVP, logically, clear correlation between the maximum dip in MAP and the maximum dip in PVP should also exist. The dip (or drop) was calculated as the difference between the initial value (or baseline) and the lowest point in the first 10 min. For this evaluation, the artificial dip in both parameters during the first minutes was taken.

When computed for all rats (*n* = 51) the lowest drop in MAP is 35% compared to its initial value. This is statistically significantly different from zero (*p* < 0.001). Differentiating among the three groups, the figures are 28.6% (group one), 39.3% (group two), and 35.8% (group three). The ANOVA indicates that the drops in MAP in groups one and two are significantly different (*p* = 0.03) as shown in the post-hoc Bonferroni adjusted pairwise comparison of means. The lowest drop in PVP for all rats (*n* = 51) is 11.8% compared to its initial value, also statistically significantly different from zero (*p* < 0.001). Differentiating among the three groups, the figures are 10.0% (group one), 14.1% (group two), and 11.1% (group three). No significant differences in the drop in PVP between the three groups is found (Table 3).

The scatter plot of % maximal drop in MAP vs. % maximal drop in PVP shows the significant correlation (r = 0.8173), further suggesting a close correlation between MAP and PVP (Figure 2).

### 2.5. Quantifying the Time Lag between Pressure Change in MAP and PVP during the First 10 min

This type of evaluation was aimed at identifying whether a change in MAP induces a change in PVP or vice versa. For each animal, the time point of minimal pressure in MAP and PVP was determined. For quantifying the time delay between the lowest drop in MAP and PVP, the sample size was trimmed at 95% percentile.

Based on the *t*-test computed for all groups, there is a significant positive lag (*p* < 0.001) between minimal pressure of MAP and minimal pressure of PVP with a mean of +24 s. These data suggest a cause-and-effect relationship of drop in MAP and drop in PVP. Figure 3 is the graphical representation of the estimated time distribution showing the distribution of mean time delay between the lowest drop in MAP and lowest drop in PVP in all rats (*n* = 51).

### 2.6. Elevation of MAP and Its Effect on PVP

During the preparation phase, significant bleeding occurred in a few animals. These were not included in the analyses. Norepinephrine was used to stabilize the circulation. Monitoring of MAP and PVP shows an immediate increase in arterial pressure. After a time delay of approximately 20 s, there is an increase in PVP (Figure 4). These data suggest that an increase in MAP is followed by an increase in PVP after a certain time lag.

## 3. Discussion

To date, medical therapy of portal hypertension includes vasoactive substances, such as NSBB (propranolol, carvedilol) or organic nitrates, among others. Therefore, it may be surmised that a drug that lowers systemic blood pressure would also influence the portal circulation [9,17]. However, experimental or clinical studies unequivocally proving a direct correlation are sparse.

In this study, we manipulated MAP in rats with healthy livers by bolus injection of 600 µL fluid and monitored MAP and PVP up to 30 min after injection. In the control (group one), 0.9% saline was injected. As pharmacological intervention with a vasoactive drug, the PDE-5-inhibitor sildenafil was used in two doses (low-dose sildenafil (group two), 0.1 mg/kg in saline, high-dose Sildenafil (group three) 1 mg/kg in saline). Similar to a previous study [45], the bolus injection of fluids induces a reversible cardiac decompensation due to volume overload lasting less than 10 min. The latter may be explained by the fact that the injected volume exceeds the stroke volume of the rat heart (60 µL) ten-fold.

The main findings of this study are as follows:

Considering the courses of MAP and PVP in the time interval from 10 to 30 min (i.e., after cardiac restabilization) the correlation between MAP and PVP is statistically evaluated. For every 1% change in MAP, PVP changes by 0.48% in group one, suggesting a close correlation between the two pressure values. For every 1% change in MAP, PVP changes by 0.55% in group two, and by 0.65% in group three. There is a significant difference between groups one and two, one and three, and two and three. This suggests that sildenafil exerts a dose-dependent effect on PVP beyond its effect on MAP. Sildenafil’s action becomes evident only several minutes after injection, which is consistent with previous observations [45] and may be explained by its complex molecular action [46,47].

The courses of MAP and PVP during the first 10 min (i.e., during the circulatory destabilization) were evaluated in terms of two aspects:There is high correlation between the maximal drop in MAP and the maximal drop in PVP. This further suggests a close relationship between the two circulatory systems;There is a distinct time lag between the lowest drop in MAP and the lowest drop in PVP. It totals up to 24 s as calculated for all groups (*p* = 0.0001). The difference among the groups is not statistically significant. These data imply a cause-and-effect relation between the drop in MAP and drop in PVP.

An injection of norepinephrine induces a sharp rise in MAP followed by a rise in PVP after a short time lag. Although this was an observation in only three rats in which surgically induced blood loss led to cardiac depression, the effect was striking enough to show that a change in MAP influenced PVP in both directions. A decrease in MAP induces a decrease in PVP, while an increase in MAP induces an increase in PVP.

This study provides additional evidence on the question of whether or not a medical therapy affecting systemic blood pressure also has an effect on portal pressure. It shows that there is a close relationship between change in systemic pressure and change in portal pressure, at least in healthy livers. Furthermore, the study suggests an effect of sildenafil on portal pressure that exceeds the effect on systemic blood pressure.

It should be stressed that the present data were obtained in animals with healthy livers. It is of utmost clinical interest whether these effects could also be reproduced in livers with cirrhosis. In such a complex disease process, the hemodynamic regulation is unfavorably altered. Intra-sinusoidal upregulation of phosphodiesterase-5 (PDE-5) leads to decreased cyclic guanosine monophosphate (cGMP) and, consequently, to a constriction of hepatic sinusoids followed by impaired liver function and elevation in portal pressure [27]. In the periphery, PDE-5 is downregulated in progressed liver cirrhosis, leading to a hyperdynamic state. The use of PDE5-inhibitors in liver cirrhosis is currently being investigated in clinical trials. To better interpret the findings and changes in liver cirrhosis, one has to understand the physiological regulation in healthy animals.

Based on several studies, it may be inferred that a correlation between systemic blood pressure and the extent of portal hypertension is probable.

In one of the few similar animal studies, Kravetz et al. [43] measured the effect of blood re-transfusion after an induced hemorrhage of approximately 17% of blood volume in rats with CCl_4_-induced cirrhosis. In rats with high porto-systemic shunts, blood re-transfusion increased portal pressure. However, a potential change in systemic hemodynamic parameters was not considered.

Similarly, Castañeda et al. [44] investigated the effect of bleeding and re-transfusion in BDL rats with liver cirrhosis and measured systemic and portal hemodynamic parameters. Bleeding induced a drop in MAP by 44% and PVP by 46%. Re-transfusion partly restored the drop in MAP and increased PVP again. Their results suggest a correlation between MAP and PVP. However, equivoluminal re-transfusion induced a rebleeding with a drop in MAP by 12% and a drop in PVP by 5%. These and similar results finally led to the recommendation that in esophageal variceal bleeding, the transfusion of packed red blood cells should not increase the Hb content to more than 7–8 g/dL [8].

In a clinical study of patients who elected for a surgical shunt procedure, an expansion of plasma volume by dextran in 0.9% saline resulted in an increase in portal pressure and inferior vena cava pressure. However, systemic hemodynamic parameters were not reported [48].

Nearly 20 years later, a study investigated the effect of sedation in patients with liver cirrhosis [49]. Sedation with propofol induced a drop in PVP simultaneous to a drop in systemic blood pressure. Besides demonstrating the correlation between MAP and PVP, this study shows that even minimal sedation might lead to an underestimation of HVPG in the human setting. This and similar observations led to the recommendation that HVPG measurement should be performed under minimal sedation [8].

Based on available key studies on medical therapy of portal hypertension, a correlation between systemic and portal pressure can be indirectly inferred. However, this aspect was not given enough weight.

Non-selective beta-blockers (NSBB, e.g., propranolol, nadolol, timolol) are the mainstay of therapy of portal hypertension [28,50,51,52,53]. These have two mechanisms of action. They lower the cardiac output via beta-1-blockade and reduce the splanchnic perfusion via beta-2-blockade, while unopposed alpha1-adrenergic activity remains constant. Beta-1-selective beta-blockers (e.g., metoprolol, atenolol) are less effective on portal pressure than NSBB [50,54,55]. They lower portal pressure solely via their effect on MAP. ∆MAP due to propranolol ranged between 5% and 11% and ∆PVP ranged between 10% and 23% [56]. ∆MAP induced by carvedilol, a NSBB with an additional alpha-1-blocking effect, was between 11% and 17% and ∆PVP between 19% and 28%. This drug lowers PVP more effectively than propranolol but the effect on MAP is also more pronounced [57,58,59]. Thus, effective medical therapy of portal hypertension (HVPG < 12 mm Hg, decreased by ≥20% of baseline) using an NSBB can be achieved only at the expense of a decrease in MAP. These observations were eventually integrated in the “window hypothesis”, which states that NSBBs have beneficial effects only within a narrow therapeutic window in patients with liver cirrhosis [8,50,60,61].

Albillos et al. showed that prazosin, an alpha-1-receptor blocker, reduced PVP by 25.7% while reducing MAP by 25% after acute administration [62]. PVP was reduced by 19.1% and MAP by 9.5% after chronic administration [63]. From these data, it may be deduced that the PVP-lowering effect of alpha-1-receptor blockers is, at least partly, due to its lowering effect on MAP.

Organic nitrates induce vasodilation due to NO liberation in the peripheral vessels and sinusoids. In clinical application, the decrease in portal pressure and systemic blood pressure were simultaneous, e.g., applying ISDN or ISMN in patients with liver cirrhosis lowered MAP by 8–21% and PVP by 10–20% [64,65]. These results suggest that organic nitrates reduce PVP by a direct effect on sinusoids and by lowering MAP. For this reason, the Baveno VII conference recommended close monitoring of systemic circulation whenever organic nitrates are used for therapy of PH [8].

Angiotensin II receptor blockers and angiotensin-converting enzyme inhibitors target the renin–angiotensin–aldosterone system. The effect in child A cirrhosis ranged from no effect [66] to approximately 20% decrease in portal hypertension [67,68]. In advanced cirrhosis of stage child B/C, these drugs had no effect on portal pressure. The major adverse effect is arterial hypotension, particularly in patients with high baseline serum renin concentration. As with other vasoactive drugs, the effect on portal pressure is, at least partly, secondary to the decrease in MAP [67].

Statins, inhibitors of the key enzyme of cholesterol synthesis, enhance activity of endothelial NO synthase, mediated by an effect on the Rho/Rho-kinase/Akt protein phosphorylation pathway [46,69,70,71,72]. Therefore, statins address the NO-mediated sinusoidal dilation, thus, exerting some liver-specific effect. Simvastatin significantly lowered elevated portal pressure in liver cirrhosis, while MAP was unaffected [73]. However, the effect of simvastatin in the clinical setting was only modest ([74]).

Phosphodiesterase-5-inhibitors (PDE-5-I) inhibit the conversion of cGMP (which induces the dilation of smooth vascular muscle cells and hepatic stellate cells) to 5′-GMP. For several years, studies with PDE-5-Is yielded contradictory results [75,76]. Later on, PDE-5-I were found to decrease sinusoidal resistance by dilation of the constricted sinusoids in cirrhotic liver [32,45,77]. Schaffner et al. demonstrated that sildenafil in the TAA-induced rat liver cirrhosis lowered portal pressure and illustrated the biochemical background of this effect. NO synthesis remains constant, but the degradation of cGMP, the final vasodilator, by overexpressed PDE-5 in cirrhosis is inhibited [33]. In a clinical study, the PDE5-I udenafil lowered portal pressure in patients with child A/B cirrhosis in a dose-dependent manner. The portal pressure decreased by approximately 20%, while MAP was lowered by only 5% [34]. Recently, these findings were supported by Trebicka et al. [78,79]. The effect of PDE-5-Is on MAP was much less than its effects on portal pressure. This suggests that specifically targeting the intra-sinusoidal NO-cGMP pathway treatment of portal hypertension may be possible without significantly affecting MAP.

The study has some limitations. The data were collected in animals with healthy livers. The deep sedation might have lowered systemic arterial blood pressure and portal venous blood pressure. Therefore, the relationship of MAP and PVP could be confounded by the anesthesia. In the present study, some animals with TAA-induced liver cirrhosis were investigated as well. However, the heterogeneity of cirrhosis state made the statistical evaluation unreliable. Furthermore, no data on intrahepatic resistance, portal flow, or cardiac index were recorded. For a much broader understanding of portal hemodynamics, these parameters should also be considered for future research. Such studies are needed to further investigate the relationship between systemic arterial pressure and portal venous pressure in the context of liver cirrhosis.

## 4. Materials and Methods

### 4.1. Animal Preparation and Ethics Statement

Adult male Sprague Dawley rats (*n* = 55, weighing 320–420 g) were obtained (Charles River, Germany) and housed in the animal facility of the University Hospital of Freiburg with 12 h light and dark cycles and an ambient temperature of 22–25 °C. The animals were kept in individually ventilated cages with free access to food and water and were allowed to acclimatize for an average of one and a half weeks before the experiment. The experimental protocol was approved by Freiburg Animal Care and Use Committee (Regierungspräsidium Freiburg, ref.no.G13/89). All animals received care in compliance with the rules and regulations of the German Animal Protection Law and the European Animal Care Guidelines (2010/63/EU). All experimental procedures were performed under anesthesia and all efforts were made to minimize suffering.

### 4.2. Experimental Protocol

Fifty-five rats were divided into three groups. The first group (*n* = 19) received 600 µL saline solution (0.9% NaCl), while the second group (*n* = 18) was given 0.1 mg/kg sildenafil citrate (Revatio^®^ Pfizer, Freiburg, Germany) and last group (*n* = 18) received 1 mg/kg sildenafil citrate, dissolved in 600 µL saline solution (0.9% NaCl). The animals were observed daily during the period of treatment for signs of stress such as decreased activity and reduced water and food consumption.

After one and a half hours of fasting, each animal was anesthetized with inhalational isoflurane and ~100 mg/kg pentobarbital intraperitoneally. Adequate anesthesia was monitored by the withdrawal response to paw pinch. Tracheostomy was performed and continuous oxygen flow (~1 L/min) was administered via mechanical ventilation. The right external jugular vein was cannulated using two catheters (polyethylene-10) that were advanced up to the right atrium. One catheter was used for the administration of fluids and drugs and the other for monitoring of the central venous pressure (CVP). An additional polyethylene-50 catheter was inserted into the left carotid artery for measuring the MAP and systemic blood pressure.

For muscular relaxation, each animal was given 0.4 mg/mL pancuronium intraperitoneally. After performing median laparotomy, the portal vein was isolated and a flow probe (Transonic T206 flow meter) was attached, recording the portal flow rate for ~5 min. A microvascular flow probe (DP8C laser Doppler monitor DRT4) was placed on a defined position on the surface of the right liver lobe to measure the parenchymal blood flow. The portal vein was later cannulated (26G Terumo^®^, Terumo, Eschborn, Germany, cannula) to measure the PVP. This cannula was fixed using cyanoacrylate glue. Body temperature was maintained at 37–37.5 °C using a heated operating table. Oxygen saturation was continuously monitored with a pulse oximeter.

To compensate for evaporative losses during surgery, crystalloid solution (Jonosteril^®^ Bad Homburg, Germany) with 15 mg/mL pentobarbital was continuously infused at a flow rate of 1 mL/h via right jugular vein. Vital parameters were recorded throughout the whole procedure. The hepatic perfusion parameters of interest consisted of PVP, hepatic microvascular flow, and portal flow rate. Systemic hemodynamic parameters included CVP and MAP.

### 4.3. Data Processing and Statistical Analysis

All data were recorded and stored using the monitoring program basic data acquisition software version 1.5 (Hugo Sachs Electronik-Harvard Apparatus GmbH, Germany). Information such as animal identification number, weight, TimeStamp, Marker, MAPmean, and PVPmean were extracted from the database and exported to Microsoft Excel for initial data processing and management. Statistical analysis was performed using SPSS software (Version 29), STATA (Verson 17), and R software (Version 4.22). Means and standard deviations were calculated to describe the study variables. Comparison of means using analysis of variance and correlation tests were applied where appropriate. The significant level was set to alpha (α) = 0.05.

## 5. Conclusions

This study shows a close relationship between systemic arterial pressure and portal venous pressure in an animal model with healthy liver: a decrease/increase in mean arterial pressure has a corresponding decrease/increase in portal venous pressure. In the evaluation of a vasoactive drug for therapy of portal hypertension, it must be considered whether its effect is only due to a lowering of systemic blood pressure or if it exerts a specific effect on portal hemodynamics beyond its action on systemic blood pressure. This study suggests that inhibitors of PDE-5, addressing the NO–cGMP signal system, might have positive effects on portal pressure that exceed its lowering effect on systemic blood pressure.

## Figures and Tables

**Figure 1 ijms-24-09963-f001:**
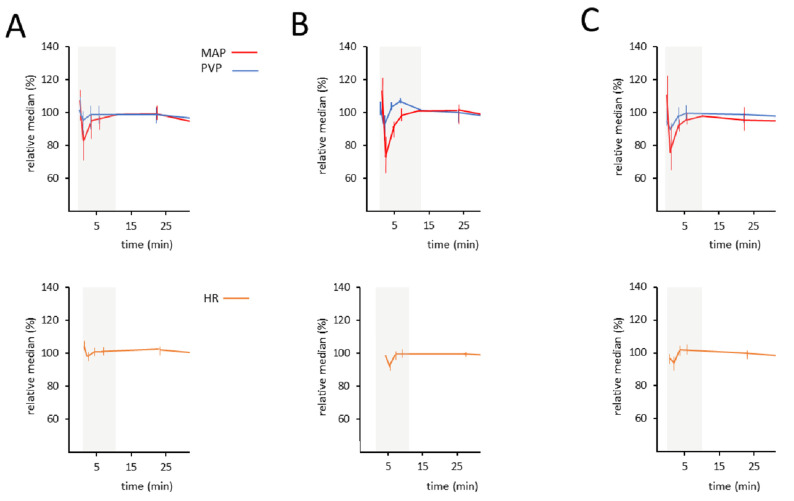
Course of MAP, PVP, and heart rate in healthy rats over time. (**A**) Group one (NaCl 0.9%). (**B**) Group two (sildenafil 0.1 mg/kg). (**C**) Group three (sildenafil 1 mg/kg). Relative change to median normalized to 100%. Bars reflect the 95% confidence interval.

**Figure 2 ijms-24-09963-f002:**
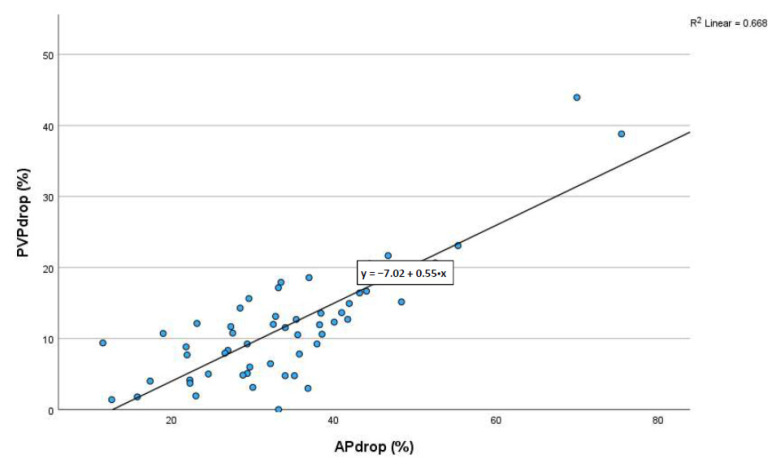
Scatter plot of % maximal drop in MAP vs. maximal drop in PVP.

**Figure 3 ijms-24-09963-f003:**
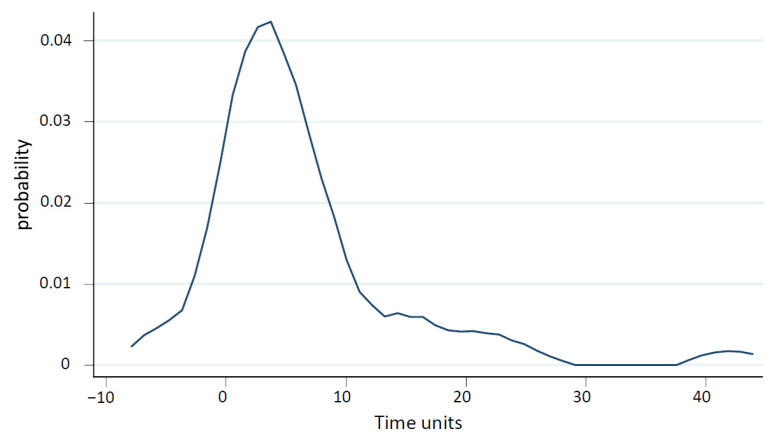
Distribution of mean time delay between the lowest drop in MAP and lowest drop in PVP in all rats. Each group has a significant positive lag (all *p*-values < 0.05) with mean values of 30 s for group one (*p* = 0.0122), 22 s for group two (*p* = 0.0034), and 20 s for group three (*p* = 0.0081). The difference between those estimates is not significant. One time unit corresponds to two seconds.

**Figure 4 ijms-24-09963-f004:**
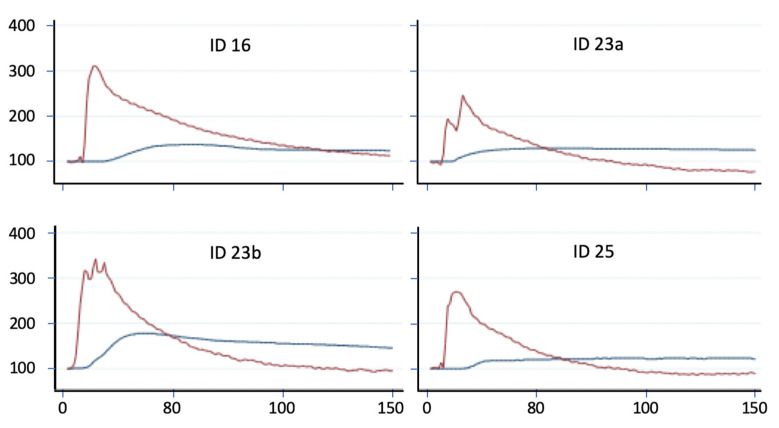
Effects of norepinephrine on MAP and PVP in three rats with intraprocedural bleeding. Rat with ID23 received norepinephrine twice. Pressure measurements are normalized to 100%. 
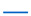
 PVPmean, 
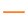
 MAPmean.

**Table 1 ijms-24-09963-t001:** Baseline characteristics of rats by treatment groups.

	Group One NaCl 0.9% *n* = 15	Group Two Sildenafil (0.1 mg/kg) *n* = 18	Group Three Sildenafil (1 mg/kg) *n* = 18
Weight (g)	375.25 ± 14.60	371.55 ± 20.42	377.72 ± 18.99
MAP mean (mmHg)	85.55 ± 13.79	87.20 ± 15.02	79.69 ± 13.75
PVP mean (mmHg)	6.42 ± 0.59	6.59 ± 0.55	6.26 ± 0.63

**Table 2 ijms-24-09963-t002:** Regression analyses between MAP and PVP in different groups of healthy rats, calculated over time interval 10–30 min after injection of fluid. 95% confidence intervals in brackets.

	Evaluation Time Interval 10–30 min
Observed time	10 min–30 min
Group one (NaCl 0.9%)	0.485 **^, †^ [0.464, 0.505]
Group two (sildenafil 0.1 mg/kg)	0.550 ***^, †^ [0.541, 0.560]
Group three (sildenafil 1 mg/kg)	0.651 ***^, †^ [0.632, 0.670]

Entries in bold mean: % change (increase or decrease) in PVP for every 1% change (increase or decrease) of MAP. ** *p* < 0.01, *** *p* < 0.001. Pairwise comparison of coefficients with *t*-test between Group 1 and 2, Group 1 and 3 and Group 2 and 3: ^†^
*p* < 0.001.

**Table 3 ijms-24-09963-t003:** Mean values of MAP drop and PVP drop (%). 95% confidence intervals in brackets.

Group	*n*	Mean MAP Drop (%)	Mean PVP Drop (%)
Total	51	35.0 (31.6, 38,3)	11.8 (9.4, 14.1)
Group one (NaCl)	15	28.6 ^†^ (23.3, 34.0)	10.0 (6.6, 13.4)
Group two (sildenafil 0.1 mg/kg)	18	39.3 ^†^ (33.1, 45.5)	14.1 (9.8, 18.3)
Group three (sildenafil 1 mg/kg)	18	35.8 (30.2, 41.5)	11.1 (6.2, 16.0)

^†^ significant difference between group 1 and group 2 *p* = 0.03.

## Data Availability

Not applicable.

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
