# Peer review of "Close Relationship between Systemic Arterial and Portal Venous Pressure in an Animal Model with Healthy Liver"

_ijms, 2023, doi:10.3390/ijms24129963_

Round 1
Reviewer 1 Report
The study examines the correlation between changes in MAP and PP after administering iv: a) 0.6 ml of saline, b) sildenafil low dose (in the same volume), or c) sildenafil high dose to anesthesized, mechanically ventilated and laparotomized healthy rats.
Injection of saline caused a sharp increase in arterial pressure followed by a deep hypotension that recovered almost completely after 10 min, followed by a gradual, mild decrease in MAP. During the initial decrease of MAP, there was a much less marked decrease in PP. After 10 min, PP did not decreased at all (but MAP did).
Behaviour of MAP and PP after sildenafil (low and high doses) was qualitatively similar, but the changes in MAP were exaggerated as compared with rats receiving saline. After 10 min, MAP and PP decreased slightly. Changes in PP and in MAP in that part were almost identical.
The authors examined the correlation between changes in MAP and PP and say these to be significant, but data is presented in a way that does not follow what shown in the Figure, where it is evident that while during the first 10 min changes in MAP much exceeded those in PP, this was not the case after 10 min. Also, the greater effects after high dose than low dose are perhaps significant in the way presented, but it is very doubtful that these are of any physiological relevance.
I have many objections with this study.
Animals in the 3 groups were quite hypotensive at baseline. This is likely related with the fact that the study was done under heavy anesthesia, mechanical ventilation, and a laparotomy. This by itself markedly influences both baseline hemodynamics and may interfere with the response to certain stimuli (i.e., adrenergic stimulation).
The study findings are used to question the use of vasoactive agents for portal hypertension and suggest that the effects could be predicted from changes in MAP. This speculation is by no way proved by the study and is in contradiction with the very well documented fact that changes in portal pressure gradient (or in HVPG) in patients with cirrhosis and portal hypertension after for instance propranolol do not hold any relationship with changes in MAP. The reasoning could be even dangerous, since may suggest that the greater the effect on MAP, the better for the patient since this will guarantee a more profound decrease in portal pressure. Actually, it is recommended to stop portal hypertensive medication in case of systemic hypotension.
Hemodynamic changes can not be interpreted from only measurements of blood pressure, without simultaneous measurements of blood flow. This will assume that blood flow is constant during all the study and that changes in pressure reflect merely changes in vascular resistance. This does not occur in vivo. It is surprising that we are not given data on blood flow when this was actually measured, according with what stated in methods.
To derive conclusions on what can happen in portal hypertension from studies in healthy animals is totally inadequate and a fundamental mistake.
Author Response
Open Review
(x) I would not like to sign my review report
( ) I would like to sign my review report
Quality of English Language
( ) English very difficult to understand/incomprehensible
( ) Extensive editing of English language and style required
( ) Moderate English changes required
(x) English language and style are fine/minor spell check required
( ) I am not qualified to assess the quality of English in this paper
|
Yes |
Can be improved |
Must be improved |
Not applicable |
|
|
Does the introduction provide sufficient background and include all relevant references? |
( ) |
( ) |
(x) |
( ) |
|
Are all the cited references relevant to the research? |
( ) |
(x) |
( ) |
( ) |
|
Is the research design appropriate? |
( ) |
( ) |
(x) |
( ) |
|
Are the methods adequately described? |
( ) |
(x) |
( ) |
( ) |
|
Are the results clearly presented? |
( ) |
( ) |
(x) |
( ) |
|
Are the conclusions supported by the results? |
( ) |
( ) |
(x) |
( ) |
Comments and Suggestions for Authors
The study examines the correlation between changes in MAP and PP after administering iv: a) 0.6 ml of saline, b) sildenafil low dose (in the same volume), or c) sildenafil high dose to anesthesized, mechanically ventilated and laparotomized healthy rats.
Injection of saline caused a sharp increase in arterial pressure followed by a deep hypotension that recovered almost completely after 10 min, followed by a gradual, mild decrease in MAP. During the initial decrease of MAP, there was a much less marked decrease in PP. After 10 min, PP did not decreased at all (but MAP did).
Behaviour of MAP and PP after sildenafil (low and high doses) was qualitatively similar, but the changes in MAP were exaggerated as compared with rats receiving saline. After 10 min, MAP and PP decreased slightly. Changes in PP and in MAP in that part were almost identical.
The authors examined the correlation between changes in MAP and PP and say these to be significant, but data is presented in a way that does not follow what shown in the Figure, where it is evident that while during the first 10 min changes in MAP much exceeded those in PP, this was not the case after 10 min. Also, the greater effects after high dose than low dose are perhaps significant in the way presented, but it is very doubtful that these are of any physiological relevance.
Answer: Thank you for this comment. In the revised version, it is now emphasized that the first 10 minutes reflected the artificial deterioration of MAP caused by fluid overload. This time period was used to show the link between MAP and PVP, as well as to demonstrate the time lag in which the PVP dropped after the MAP.
I have many objections with this study.
Animals in the 3 groups were quite hypotensive at baseline. This is likely related with the fact that the study was done under heavy anesthesia, mechanical ventilation, and a laparotomy. This by itself markedly influences both baseline hemodynamics and may interfere with the response to certain stimuli (i.e., adrenergic stimulation).
Answer: Thank you for this remark. This concern is now mentioned in the Discussion part.
The study findings are used to question the use of vasoactive agents for portal hypertension and suggest that the effects could be predicted from changes in MAP. This speculation is by no way proved by the study and is in contradiction with the very well documented fact that changes in portal pressure gradient (or in HVPG) in patients with cirrhosis and portal hypertension after for instance propranolol do not hold any relationship with changes in MAP. The reasoning could be even dangerous, since may suggest that the greater the effect on MAP, the better for the patient since this will guarantee a more profound decrease in portal pressure. Actually, it is recommended to stop portal hypertensive medication in case of systemic hypotension.
Answer: It is not our goal to question the use of vasoactive agents. In this study, we emphasized that lowering of MAP will lower PVP to a certain extent. This has to be kept in mind when assessing the efficacy of a pharmacological substance on PVP. This is now mentioned in the Introduction and further explained in the Discussion.
Hemodynamic changes can not be interpreted from only measurements of blood pressure, without simultaneous measurements of blood flow. This will assume that blood flow is constant during all the study and that changes in pressure reflect merely changes in vascular resistance. This does not occur in vivo. It is surprising that we are not given data on blood flow when this was actually measured, according with what stated in methods.
Answer: Blood flow was initially measured in the first few animals, however, it led to a higher drop out rate due to procedural complications. The animals of Figure 3 died due to manipulation of the cuff around the portal vein. Thus, we decided to do away with flow measurements and focus on pressure instead. We supplemented the pulse trajectories in Figure1 to better evaluate the hemodynamics and regulatory processes (as also suggested by one of the reviewers).
To derive conclusions on what can happen in portal hypertension from studies in healthy animals is totally inadequate and a fundamental mistake.
Answer: We consider it necessary to explore the relationship between MAP and PVP in healthy animals to better understand the changes in portal hypertension. Only then it would be possible to assess whether the correlations of both parameters are altered in the context of portal hypertension.
Reviewer 2 Report
The work describes the relationship between portal venous pressure and mean arterial pressure in healthy rats in which hemodynamics are altered by the PDE-5 blocker sildenafil (S) (0.1mg/kg and 1mg/kg).
There is a drop of arterial pressure and portal pressure after saline (slight), the low dose of S and the high dose of S.
The reviewer has some question which should be clarified:
- The drop after saline administration are explained by transient cardial decompensation? Where do the authors get these statements from? Was the CO measured? Also, for the explanation of the other findings, one would like to have the CO data.
- With regard to Figure 1, the reviewer recommends a larger format and also an additional figure with the absolute values. This figure is the core of their findings.
- It is unclear why the MAP drop-off is greater with the lower S dose than with the high dose and why there is a rebound of PVP after 0.1mg/kg S. Is there an rapid initial increase of MAP?
- According to this figure, one has the impression that the portal pressure drop has reached its maximum even before the MAP drop and not after it (fig 2)
- Administration of S should also affect arterial blood flow to the liver. This can/could lead to a change in intrahepatic resistance. These data are needed to explain the findings, as well as the concomitant cardiac output.
- Fig 3 is interesting (as MAP increases, so does PVP). Again, how did the arterial blood flow to the liver and the intrahepatic resistance and cardiac output behave at the same time? Which unit is used for the ordinates (y-axis)?
- It would be nice to see a scatterplot (at least for the reviewer) between the delta of the absolute MAP levels (maximal decrease) against the delta of the absolute HVP levels (maximal decrease) for the two groups.
- S was probably applicated intravenously. Is this mentioned?
There is an important problem in this work. The study on the relationship between the HVP and the MAP and their modulation is actually mainly interesting in animal models with liver cirrhosis.
Author Response
Open Review
(x) I would not like to sign my review report
( ) I would like to sign my review report
Quality of English Language
( ) English very difficult to understand/incomprehensible
( ) Extensive editing of English language and style required
(x) Moderate English changes required
( ) English language and style are fine/minor spell check required
( ) I am not qualified to assess the quality of English in this paper
|
Yes |
Can be improved |
Must be improved |
Not applicable |
|
|
Does the introduction provide sufficient background and include all relevant references? |
( ) |
( ) |
(x) |
( ) |
|
Are all the cited references relevant to the research? |
(x) |
( ) |
( ) |
( ) |
|
Is the research design appropriate? |
( ) |
(x) |
( ) |
( ) |
|
Are the methods adequately described? |
( ) |
(x) |
( ) |
( ) |
|
Are the results clearly presented? |
( ) |
(x) |
( ) |
( ) |
|
Are the conclusions supported by the results? |
(x) |
( ) |
( ) |
( ) |
Comments and Suggestions for Authors
The work describes the relationship between portal venous pressure and mean arterial pressure in healthy rats in which hemodynamics are altered by the PDE-5 blocker sildenafil (S) (0.1mg/kg and 1mg/kg).
There is a drop of arterial pressure and portal pressure after saline (slight), the low dose of S and the high dose of S.
The reviewer has some question which should be clarified:
- The drop after saline administration are explained by transient cardial decompensation? Where do the authors get these statements from? Was the CO measured? Also, for the explanation of the other findings, one would like to have the CO data.
Answer: We think this drop in MAP and heart rate was due to volume overload. The reduction in heart rate occurred simultaneously with the MAP drop. We added the heart rate course in Figure 1 to elucidate these findings. It was emphasized that this was a procedural artifact. Unfortunately, the CO data are not available.
- With regard to Figure 1, the reviewer recommends a larger format and also an additional figure with the absolute values. This figure is the core of their findings.
We thank reviewer 2 for this remark. The figures are now enlargend and sharper.
- It is unclear why the MAP drop-off is greater with the lower S dose than with the high dose and why there is a rebound of PVP after 0.1mg/kg S. Is there an rapid initial increase of MAP?
Answer: As already mentioned, the drop of MAP was a procedural artifact. Most likely it occurred in the context of volume overload. A drop in MAP led to a smaller drop in PVP with time delay. However, the absolute drop in MAP may depend on time of volume administration, therefore, we omitted further analyses of these data.
- According to this figure, one has the impression that the portal pressure drop has reached its maximum even before the MAP drop and not after it (fig 2)
Answer: Our graphical output moved the MAP curves to the right. Once we corrected the output, it clearly shows that PAP followed MAP. Furthermore, the statistical analysis confirmed this (in the mentioned time span of 24 seconds as was shown in Figure 3).
- Administration of S should also affect arterial blood flow to the liver. This can/could lead to a change in intrahepatic resistance. These data are needed to explain the findings, as well as the concomitant cardiac output.
Answer: This is a reasonable suggestion. However, during the initial experiments we lost several animals due to complications caused by attaching the measuring probes to the portal vein. Thus, we did away with flow measurements. This was mentioned in the Discussion as a limitation of this study.
- Fig 3 is interesting (as MAP increases, so does PVP). Again, how did the arterial blood flow to the liver and the intrahepatic resistance and cardiac output behave at the same time? Which unit is used for the ordinates (y-axis)?
Answer: Flow data are not available, as well as measurement of intrahepatic resistance.
The legend described that pressure measurements are normalized. We added the “%” to the y-axis.
- It would be nice to see a scatterplot (at least for the reviewer) between the delta of the absolute MAP levels (maximal decrease) against the delta of the absolute HVP levels (maximal decrease) for the two groups.
Answer: Thank you for this suggestion. We now included a scatter plot which could help to better understand the link between MAP and PVP.
- S was probably applicated intravenously. Is this mentioned?
Answer: Yes, this was mentioned in the Methods section.
There is an important problem in this work. The study on the relationship between the HVP and the MAP and their modulation is actually mainly interesting in animal models with liver cirrhosis.
Answer: Although it is of utmost clinical importance to investigate the dependence of PVP on MAP in the context of portal hypertension, we believe, however, that to better interpret the findings and changes in liver cirrhosis, one has to understand the physiological regulation in healthy animals. These initial observations could be a good basis for future research using animal models with liver cirrhosis.
Reviewer 3 Report
This article explored to what extent MAP affects PVP by manipulating animal MAP through bolus injection. This article found that PVP changes lag MAP changes by an average of 24s. In addition, sildenafil could amplify the impaction of MAP on PVP, which suggests sildenafil’s effects beyond the change of MAP due to bolus injection. This study suggests that when evaluating the effect of an intervention on reducing portal pressure, attention should be paid to assess whether the effect achieves only through reducing circulating pressure, which has been neglected in many basic studies related to portal hypertension.
However, there are some problems, which would better be solved before it is considered for publication.
l The data were collected from healthy SD rats. Can these results represent the hemodynamic changes in portal hypertension situation?
l In experimental protocol, it is necessary to emphasize that different doses of sildenafil according to the body weight are all dissolved in 600μL saline, or it might be ambiguous.
l Please provide the data on heart rate changes, as the effect of vasoactive drugs on heart rate is of great significance.
l In evaluation 5, to what extent MAP affects PVP when MAP increases?
l Figure 2 shows that PVP changes earlier than MAP in several rats. How do you explain this phenomenon?
l The discussion section is too long and detailed. Many contents are not directly related to this study. Please simplify your discussion section, especially the discussion on other vasoactive drugs.
l The abbreviation of phosphodiesterase-5 is used in the second paragraph of background and aims, while the full name of phosphodiesterase-5 appears in the next paragraph, which is a little bit strange. Such kind of minor faults exist in other places.
Author Response
Open Review
( ) I would not like to sign my review report
(x) I would like to sign my review report
Quality of English Language
( ) English very difficult to understand/incomprehensible
( ) Extensive editing of English language and style required
( ) Moderate English changes required
( ) English language and style are fine/minor spell check required
(x) I am not qualified to assess the quality of English in this paper
|
Yes |
Can be improved |
Must be improved |
Not applicable |
|
|
Does the introduction provide sufficient background and include all relevant references? |
(x) |
( ) |
( ) |
( ) |
|
Are all the cited references relevant to the research? |
(x) |
( ) |
( ) |
( ) |
|
Is the research design appropriate? |
( ) |
(x) |
( ) |
( ) |
|
Are the methods adequately described? |
( ) |
(x) |
( ) |
( ) |
|
Are the results clearly presented? |
(x) |
( ) |
( ) |
( ) |
|
Are the conclusions supported by the results? |
(x) |
( ) |
( ) |
( ) |
Comments and Suggestions for Authors
This article explored to what extent MAP affects PVP by manipulating animal MAP through bolus injection. This article found that PVP changes lag MAP changes by an average of 24s. In addition, sildenafil could amplify the impaction of MAP on PVP, which suggests sildenafil’s effects beyond the change of MAP due to bolus injection. This study suggests that when evaluating the effect of an intervention on reducing portal pressure, attention should be paid to assess whether the effect achieves only through reducing circulating pressure, which has been neglected in many basic studies related to portal hypertension.
However, there are some problems, which would better be solved before it is considered for publication.
l The data were collected from healthy SD rats. Can these results represent the hemodynamic changes in portal hypertension situation?
Answer: Understanding the relationship between MAP and PAP in healthy animals provides a good foundation for future studies focusing on animals with cirrhotic portal hypertension. This argument is mentioned in the revised Introduction and Discussion.
l In experimental protocol, it is necessary to emphasize that different doses of sildenafil according to the body weight are all dissolved in 600μL saline, or it might be ambiguous.
Answer: Thank you for this remark. We have added this information in the Methods section.
l Please provide the data on heart rate changes, as the effect of vasoactive drugs on heart rate is of great significance.
Answer: Thank you for this suggestion. Figure 1 has been revised accordingly.
l In evaluation 5, to what extent MAP affects PVP when MAP increases?
Answer: These are data from three animals which died due to circulatory problems secondary to intraprocedural bleeding. We did not include these observations in the statistical analysis and provided qualitative description instead.
l Figure 2 shows that PVP changes earlier than MAP in several rats. How do you explain this phenomenon?
Answer: This phenomenon is a software peculiarity such that the error bars do not lie on top of each other. We changed the output in the revised version and Figure 1 now clearly shows that MAP dip preceded the PVP dip.
l The discussion section is too long and detailed. Many contents are not directly related to this study. Please simplify your discussion section, especially the discussion on other vasoactive drugs.
Answer: The Discussion part was shortened and simplified as suggested.
l The abbreviation of phosphodiesterase-5 is used in the second paragraph of background and aims, while the full name of phosphodiesterase-5 appears in the next paragraph, which is a little bit strange. Such kind of minor faults exist in other places.
Answer: Thank you for this comment. Spellings have been corrected.
Reviewer 4 Report
the topic is very interesting and well presented.
the statistical analysis is a bit complex and especially the comparison of coefficients obtained by means of the t-test is difficult to understand
Author Response
Open Review
( ) I would not like to sign my review report
(x) I would like to sign my review report
Quality of English Language
( ) English very difficult to understand/incomprehensible
( ) Extensive editing of English language and style required
( ) Moderate English changes required
(x) English language and style are fine/minor spell check required
( ) I am not qualified to assess the quality of English in this paper
|
Yes |
Can be improved |
Must be improved |
Not applicable |
|
|
Does the introduction provide sufficient background and include all relevant references? |
(x) |
( ) |
( ) |
( ) |
|
Are all the cited references relevant to the research? |
( ) |
( ) |
( ) |
( ) |
|
Is the research design appropriate? |
( ) |
(x) |
( ) |
( ) |
|
Are the methods adequately described? |
( ) |
(x) |
( ) |
( ) |
|
Are the results clearly presented? |
( ) |
(x) |
( ) |
( ) |
|
Are the conclusions supported by the results? |
( ) |
(x) |
( ) |
( ) |
Comments and Suggestions for Authors
the topic is very interesting and well presented.
the statistical analysis is a bit complex and especially the comparison of coefficients obtained by means of the t-test is difficult to understand
Answer: Thank you for you helpful comments.
Round 2
Reviewer 1 Report
The authors have not addressed the questions raised during the review, but done minimal cosmetic changes and disregarded most major objections.
In my opinion the paper has not been revised appropriately.
Author Response
Answer to Reviewer 1
We had significantly changed the original manuscript and had added several parts of the manuscript according to the suggestions. These changes have been marked in red, new changes (round 2) are marked in blue.
E.g.: The Abstract was complete rewritten.
In the Background and Aims section significant parts were rewritten in order to better explain the aims of the study.
The evaluation of the courses was limited to a period of 10-30 minutes in order to exclude the artefact of the dip from the calculation. The first 10 minutes were only evaluated in terms of two aspects:
- There is high correlation between the maximal drop of MAP and the maximal drop of PVP. This further suggests a close relationship between the two circulatory systems.
- There is a distinct time lag between the lowest drop in MAP and the lowest drop in PVP. It totaled up to 24 seconds as calculated for all groups (p=0.0001).
For a better demonstration of the correlation between MAP and PVP we included a scatter plot.
We emphasized several times that the results of the study refer to the situation in healthy rats. It is necessary to perform similar studies in animals with cirrhotic liver. The present study may be the basis to interpret hemodynamic in cirrhosis. It may be taken into account that any vasoactive drug may lower MAP AND PVP. We will not overinterprete the results of the present study.
Reviewer 2 Report
It is unclear to what extent systemic arterial blood pressure influences portal pressure. This relationship is clinically important as drugs, which are conventionally used for therapy of por-tal hypertension, may also influence systemic arterial blood pressure (abstract)
- Actually, the authors' approach was different. They investigated whether changes in the systematic circulation influence portal pressure. So the question would be whether drugs that influence arterial blood pressure also lead to a change in portal pressure.
- The authors show, that there is a relationship between systemic arterial presure and portal pressure. But one should like to have more insight. I still do not unterstand why volume expansion (saline) leeds to a reduction of portal and arterial pressure. Is this a common phenomenon in rats? Then give respetive references.
- In the last paragraph on limitations the authors do not mention that they have no data on CI and intrahepatic resistance.
Author Response
Thank you for your comments. We feel, that your questions can be answered as follows:
It is unclear to what extent systemic arterial blood pressure influences portal pressure. This relationship is clinically important as drugs, which are conventionally used for therapy of por-tal hypertension, may also influence systemic arterial blood pressure (abstract)
- Actually, the authors' approach was different. They investigated whether changes in the systematic circulation influence portal pressure. So the question would be whether drugs that influence arterial blood pressure also lead to a change in portal pressure.
Answer: The dependence of portal venous pressure on systemic arterial pressure has been insufficiently studied. This is of importance insofar as the drug-induced reduction of portal venous pressure is mostly accompanied by a reduction in arterial pressure. All medications being used to lower portal pressure have a reduction of the systemic pressure as a „side effect“. In our study we show, how changes of arterial pressure influence the portal pressure: with a time lag, a pressure loss in the arterial system is also reflected to a lesser extent in the portal system. So for all drugs used to lower portal pressure, efficacy must be assessed taking into account arterial pressure reduction.
As up to date there is no medication that is specifically lowering portal pressure without changing MAP, a comparison to the sole effectiveness on the portal system is not possible.
- The authors show, that there is a relationship between systemic arterial presure and portal pressure. But one should like to have more insight. I still do not unterstand why volume expansion (saline) leeds to a reduction of portal and arterial pressure. Is this a common phenomenon in rats? Then give respetive references.
Answer: We feel, that there is no better explanation than volume overload. The heart of an adult mouse typically weighs between 100 and 200 milligrams, with the left ventricle comprising a significant portion of the total heart mass. The stroke volume will be about 60 microliters. The injection of 600 microliters within a few seconds in the right atrium will lead to a massive volume overload of the heart. According to the helpful advice of reviewer 2 we added the heart rate diagrams, that show, that also a dip in heart rate was seen. In the Discussion section, we included a phrase, which explained the situation. In another setting this phenomenon is described before (already cited as Reference 46 in the Discussion section). In this study, the volume was given over 90 seconds and into the tail vein. A dip in MAP was nevertheless observed.
- In the last paragraph on limitations the authors do not mention that they have no data on CI and intrahepatic resistance.
Answer: We have included the following phrases in the limitations.
Furthermore, no data on intrahepatic resistance, portal flow, or cardiac index were recorded. For a deeper understanding these parameter should be measured in further studies as well.
New changes in the manuscript are marked in blue.